# Multiomics biomarkers were not superior to clinical variables for pan-cancer screening

Martin Smelik[1], Yelin Zhao [1], Dina Mansour Aly[1], AKM Firoj Mahmud [1], Oleg Sysoev[2], Xinxiu Li [1,3] & Mikael Benson [1,3] ✉

## Abstract

**Background** Cancer screening tests are considered pivotal for early diagnosis and survival. However, the efficacy of these tests for improving survival has recently been questioned. This study aims to test if cancer screening could be improved by biomarkers in peripheral blood based on multi-omics data.

**Methods** We utilize multi-omics data from 500,000 participants in the UK Biobank. Machine learning is applied to search for proteins, metabolites, genetic variants, or clinical variables to diagnose cancers collectively and individually.

**Results** Here we show that the overall performance of the potential blood biomarkers do not outperform clinical variables for collective diagnosis. However, we observe promising results for individual cancers in close proximity to peripheral blood, with an Area Under the Curve (AUC) greater than 0.8.

**Conclusions** Our findings suggest that the identification of blood biomarkers for cancer might be complicated by variable overlap between molecular changes in tumor tissues and peripheral blood. This explanation is supported by local proteomics analyses of different tumors, which all show high AUCs, greater than 0.9. Thus, multi-omics biomarkers for the diagnosis of individual cancers may potentially be effective, but not for groups of cancers.

## Plain language summary

This study aimed to find out if we could improve cancer screening tests by looking for signs of cancer in blood samples. We used computer and mathematical models to analyze data from 500,000 people. We found that these blood tests were not better than existing methods for diagnosing multiple types of cancer at once. However, they did show promise for diagnosing individual types of cancer that are close to the bloodstream. This suggests that finding blood markers for cancer is complex and depends on how much the cancer affects the blood. These findings could help in the development of more effective tests for individual types of cancer in the future.

Early diagnosis of cancer is considered crucial for increasing survival but complicated by a lack of, or vague, symptoms. This has led to the investment of enormous resources in developing and implementing a wide variety of cancer screening tests[1–15]. In the United States alone, $40 to $80 billion are spent on cancer screening annually[16]. However, recent studies have questioned whether these tests result in increased survival[17–20]. The reasons for this include overdiagnosis and the variable morbidities and mortalities of different cancers[16].

There are established cancer screening methods using imaging technologies such as colonoscopy for colorectal cancer or mammography for breast cancers. The performance of these technologies showed substantial variations depending on the study characteristics[21–23].

A large number of articles have highlighted that a potential solution could be to find new biomarkers. For clinical tractability, these should ideally be measurable in peripheral blood Thousands of potential biomarkers for

cancer diagnosis have been proposed[4,5,19,24–42]. However, the clinical value of such biomarkers remains uncertain. The reasons may include difficulty in clinical implementation because of cost, inability to measure these biomarkers with routine methods in peripheral blood, or long turnover times[40].

Other reasons could be that biomarkers are often prioritized based on literature, clinical experience, or analyses of different omics data. This may be confounded by knowledge biases or, in the case of omics data, limited sample numbers. Another reason for the limited clinical value of biomarkers in peripheral blood is that molecular changes in blood may differ from those in local tumor tissues. The importance of such investigations can be inferred from the daunting complexity and heterogeneity of local molecular changes. A tumor may involve changes in interactions between thousands of genes across multiple cell types, not only within malignant cells. These changes can vary between patients with different cancer diagnoses, as well as between patients with the same diagnosis at

[1]Department of Clinical Science, Intervention and Technology (CLINTEC), Karolinska Institute, Stockholm, Sweden. [2]Division of Statistics and Machine Learning, Department of Computer and Information Science, Linköping University, Linköping, Sweden. [3]These authors jointly supervised this work: Xinxiu Li, Mikael Benson. ✉e-mail: mikael.benson@ki.se

different time points[6]. Taken together, the identification of biomarkers for cancer screening involves great challenges and unresolved questions that have not been systematically investigated: Which molecular type is optimal in terms of accuracy, tractability in routine clinical settings, as well as costs? How are biomarkers affected by if the control population is healthy or has comorbidities? How do blood biomarkers compare to the corresponding analytes in local tumor tissues? Previous studies have partially attempted to answer such questions, by for example, analyzing either specific cancer[43,44] types or specific diagnostic modalities such as genomic mutations[45] or metabolites[46].

Recent initiatives, such as UK Biobank (UKBB), a prospective study of over 500,000 participants, provide an opportunity to systematically analyze these questions. The UKBB allows searching for proteomic, metabolomic, and genomic blood biomarkers using substantially higher number of samples than before. Proteomic analysis of several diseases, including cancers, based on UKBB have identified promising biomarkers linked to both mortality and cancer incidence[47–51]. The limitations of these studies included that they were focused on incident cases, in other words patients diagnosed after blood sampling. These results led us to construct models to differentiate already diagnosed patients from healthy controls, as well as from patients with non-cancer diseases as controls, and to compare their performance with the predictive models based on tissue proteomics. Moreover, we systematically compared the predictive accuracy of different omics layers.

Briefly, we used machine learning to identify proteins, metabolites, or genetic variants to diagnose some of the most prevalent cancers collectively, as well as each of those cancers individually. For most cancers, we found limited diagnostic potential for all omics layers as well as clinical variables. However, for some cancers in highly vascularized organs, such as the kidney and the thyroid, a greater diagnostic potential was found as depicted by the high area under the receiver operating characteristic (ROC) curves (AUCs). This exception pointed to a potential explanation for the limited value of blood biomarkers. Since these organs are more exposed to blood than the others, this could result in greater overlap of biomarkers between local cancer cells and peripheral blood. This explanation was supported by local proteomics analyses of multiple cancers. In contrast to the variable AUCs of proteins in peripheral blood, the latter all had high AUCs.

While our results did not show that any proteomic, metabolomic or genomic biomarkers were superior to clinical variables for pan-cancer screening, it is possible that biomarkers for individual cancers may be clinically useful. We constructed a user-friendly Shiny app to search for and determine AUCs for such multiomics biomarkers in peripheral blood, as well as for proteomics from local cancer tissues: https://macd.shinyapps.io/CancerAtlas/.

## Methods

### Data source and study participants

The study involved participants from the UKBB, a comprehensive prospective cohort study comprising more than 500,000 individuals recruited in the United Kingdom[52]. Detailed information about the UKBB study is available on its official website (https://biobank.ndph.ox.ac.uk/showcase/). UKBB received ethical approval from the National Information Governance Board for Health and Social Care and the National Health Service Northwest Multi-Center Research Ethics Committee. All participants gave informed consent through electronic signatures before enrollment in the study. This research has been conducted under approved UKB Project ID 102162. The specific data fields utilized in this analysis included nuclear magnetic resonance (NMR) metabolomics, proteomics, imputed genomic data, date of recruitment, date of diagnosis, clinical variables, and diagnostic codes based on the International Classification of Diseases, 10th Edition (ICD-10). More detailed information about the UKBB cohort as well as a description of how the different omics analyses were performed can be found in Supplementary Methods 1.

### Diagnosis assessment and patient group identification

The diagnosis assessment in this study utilized the ICD-10 codes, which specifically extract information from the cancer registry (UKBB dataset field: 40006), as provided by the UKBB. We identified patients of the most prevalent cancers based on the number of UKBB proteomics samples. To increase the statistical power of the analyses, we followed the stratification of cancers provided by UKBB[53]. Patient groups were identified based on specific ICD-10 codes, including breast cancer (C50), bladder cancer (C76), colorectal cancer (C18, C20), kidney cancer (C64), leukemia (C91-C95), lymphoma (C82-C86), lung cancer (C34), melanoma (C34), ovarian cancer (C56), prostate cancer (C61), thyroid cancer (C73), uterine cancer (C54) and all cancers (C1-C97). The blood sampling was done at the baseline assessments between 2006 and 2010. The follow-up period for the patients was until October 31, 2022.

The cancer cases, meaning those patients who were diagnosed before or at the time of sampling, were identified as cancer patients. A group of healthy controls was identified, consisting of patients without any disease code. Similarly, a group of non-cancer controls was identified, consisting of patients with other diseases but without any cancer code. The MatchIt package[54] was used to match healthy controls as well as non-cancer controls cases with cancer cases based on age and sex. A detailed description of how the three groups were identified can be found in Supplementary Methods 2.

### Weighted polygenic PRS

PRS take into account the cumulative effect of many genetic variants, each contributing a small amount of overall risk for the disease. The UKBB imputed genetic data, consisting of over 90 million genetic variants, was utilized to calculate corresponding PRSs (1595 genetic variants in PRSs) for each cancer separately. Genetic variants within each PRS were pruned for linkage disequilibrium ($r^2 = 0.5$, 250-kb window in PLINK). Weighted PRSs were subsequently computed for all participants in the analysis of each cancer using the standard formula implemented in PLINK v1.9 software[55]. All PRS analyses included only participants of European origin and were adjusted for genetic principal components provided by the UKBB quality control files[56]. Additional information about PRS analyses can be found in Supplementary Methods 3.

### Clinical Proteomic Tumor Analysis Consortium (CPTAC) cohort

To analyze local tumor tissue proteomics data, we used CPTAC data from nine cancers, namely, breast cancer, colon cancer, head and neck cancer, kidney cancer, liver cancer, lung cancer, ovarian cancer, pancreatic cancer, and uterine cancer. Mass spectrometry was used to perform the proteomics analysis. The proteomic data from tumor samples and solid normal tissue samples were downloaded from the Cancer Proteomic Data Server (https://proteomic.datacommons.cancer.gov/pdc/). Since the aim was to compare proteomes in peripheral blood and local cancer tissues, proteins that were analyzed in both CPTAC and UKBB were included in the comparison for each cancer separately.

### Modeling the probability of health status and interpretation of the results

To construct the classifiers, a comprehensive pipeline was devised consisting of data cleaning, imputation, feature selection, classifier training, and result interpretation steps. Initially, we filtered out variables with more than 10% missing values, and the data were partitioned into training (70%) and testing (30%) sets. The k-nearest neighbor method[57] was then employed to impute missing values. Feature selection was accomplished using the extremely randomized trees method[58]. The variables were ranked based on their importance, and for downstream analysis, the top $N$ features ($N$ ranging from 1 to 15) were chosen. The prediction model was trained using logistic regression with a ridge (L2) penalty[59]. Tenfold cross-validation was used to optimize the L2 penalty parameter.

To evaluate the performance of the classifier, we computed the ROC curve, along with its corresponding AUC, using the hold-out testing set.

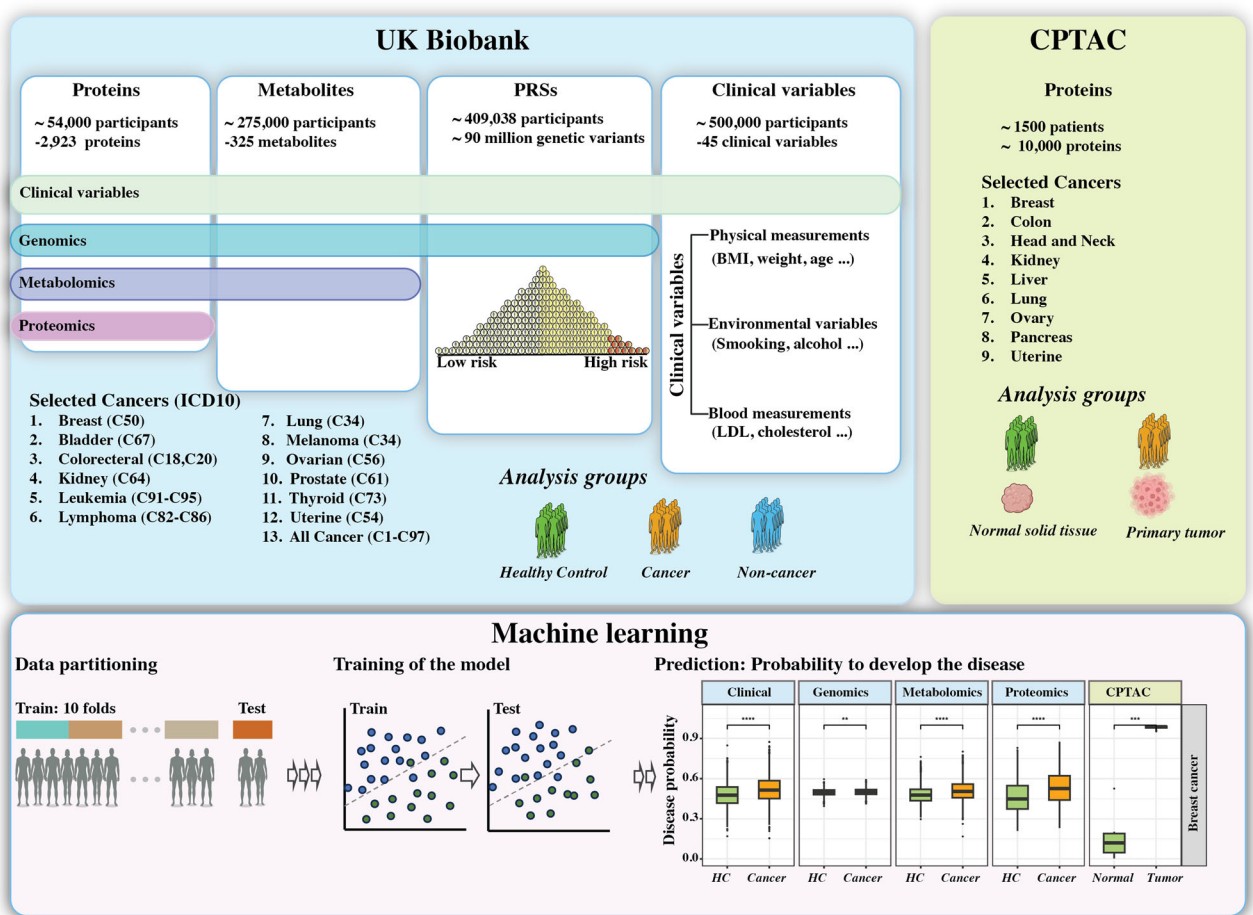

**Fig. 1 | Overview of the study.** We analyzed proteins, metabolites, genetic variants in peripheral blood, and clinical variables from cancer patients and their matched healthy controls from the UK Biobank, as well as local tumor proteomes from the Clinical Proteomic Tumor Analysis Consortium (CPTAC). Using these data, we trained prediction models to diagnose cancer patients, which were evaluated on the hold-out test dataset.

The same pipeline was applied to proteomics, metabolomics, and clinical variables from UKBB as well as for proteomics from CPTAC. For PRS, a similar pipeline was employed, omitting the feature selection step. Instead, sex, age, scaled PRS, and three genetic principal components were used as features, with the patient group as the response variable. To determine significant differences in the probability of cancer between the cancer patients and the two control groups, a two-sided $t$ test with false discovery rate adjustment was performed.

The gene ontology enrichment analysis was performed using the "enrichr" function from gseapy package, where 15 most relevant proteins for each disease and "GO_Biological_Process_2023" gene set were used as input. More details about the methodology can be found in Supplementary Methods 4.

### Construction of the interactive atlas

To allow the extraction of biomarkers for collective or individual cancer diagnosis, a user-friendly Shiny app was developed[60], accessible at https://macd.shinyapps.io/CancerAtlas/. The results presented in the atlas are derived from the previously presented predictive models. The atlas offers an interface with context-dependent adjustment options for each disease, including the type of baseline group, omics layer, and number of molecules. A more detailed description of the Shiny app is presented in Supplementary Methods 5.

The codes used in the whole manuscript are publicly available at https://github.com/SDTC-CPMed/Cancer_diagnostics.

### Inclusion and ethics statement

All collaborators of this study have been included as authors, as their participation was essential for design and implementation of the study and they fulfilled the criteria for authorship required by Nature Portfolio journals. This research was not severely restricted or prohibited in the setting of the researchers, and does not result in stigmatization, incrimination, discrimination, or personal risk to participants.

### Reporting summary

Further information on research design is available in the Nature Portfolio Reporting Summary linked to this article.

## Results

### Baseline characteristics of the study population and analyzed variables

In this study, we analyzed data from the UKBB, a repository comprising more than 500,000 participants, to investigate the efficacy of blood-based biomarkers and clinical variables for diagnosing cancer (Fig. 1). We identified 32,260 cancer patients with 86 cancers (Supplementary Data 1). Their median (IQR)] age was 62 (57–66) years, 59% of whom were female. The dataset included 2923 proteins, 325 metabolites, 90 million genetic variants, and 45 clinical variables (Supplementary Data 2).

Using machine learning, we first identified the potential diagnostic biomarkers using these variables collectively in all cancers (ICD-10 code starting with C) available in the UKBB cohort compared to controls. Next, we identified the biomarkers for individual cancers. Two distinct control

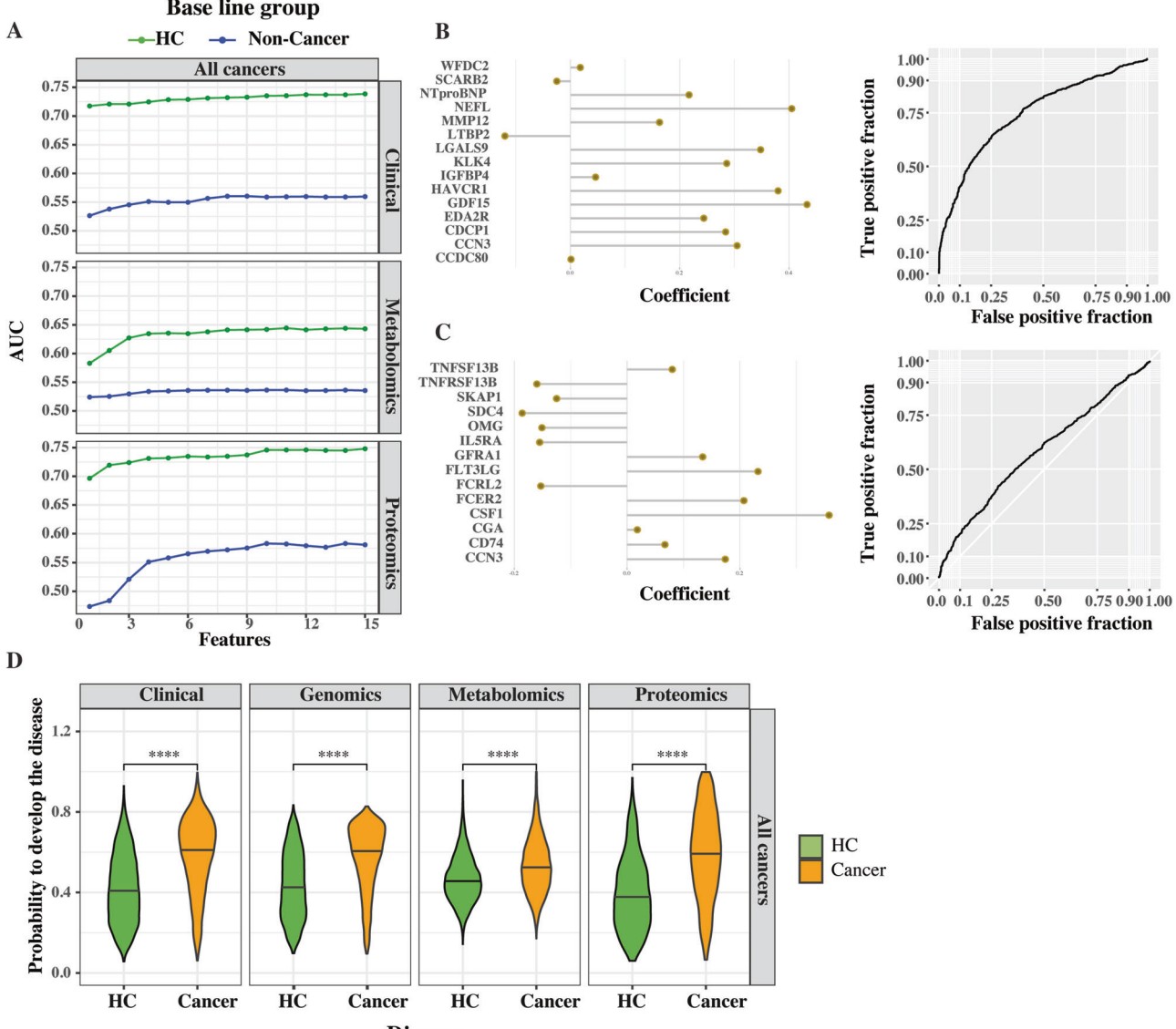

**Fig. 2 | Potential of multiomics and clinical variables for pan-cancer diagnosis.**
**A** A line plot showing the AUCs for pan-cancer biomarkers. The color indicates which group was used as a baseline control. **B, C** A dot plot depicting the protein coefficients of the proteomics model and its corresponding receiver operating characteristic curve. Positive coefficients may indicate pathogenic roles, while negative coefficients may indicate protective roles. The baseline group corresponds to healthy controls (**B**) and non-cancer patients (**C**), respectively. **D** A violin plot showing the probability of developing cancer. The line depicts the median value. ****$P < 0.0001$. T test, adjusted $P$ values.

groups, individuals without diagnosed disease and individuals with diagnosed disease other than cancer (henceforth referred to as healthy controls and non-cancer controls, respectively), were employed for baseline comparisons. The reason for including the latter control group is that comorbidities other than cancer are common. Thus, such comorbidities could potentially confound biomarker analyses. The follow-up period of the patients was until the October 31, 2022, thus the median (IQR) duration of follow-up was 5007 (481) days. Supplementary Data 3 summarizes the baseline characteristics of the cohorts.

**Potential pan-cancer biomarkers identified from multiomics data did not outperform clinical variables**
We used machine learning to analyze the efficacy of potential biomarkers from multiomics and clinical data. Specifically, we used the extremely randomized trees algorithm for feature selection followed by logistic regression with a ridge penalty using up to 15 most important features ("Methods"). Using this approach, we prioritized the most discriminating

proteins, metabolites or clinical variables associated with the cancers as a group (Fig. 2). Since we used already published polygenic risk scores (PRS), feature selection was not applicable in case of genomics. Instead, we directly applied logistic regression with a ridge penalty to diagnose the disease based on the previously published scaled PRS. We included age, sex, and three genetic principal components in the model to limit the effects of possible disease-unrelated confounders.

We used ROC curve as well as AUC values to evaluate the comparisons (shiny app, Supplementary Data 4, 5). For each cancer and omics, the highest achieved AUCs were reported. The values of AUCs above 0.8 have been suggested to be of clinical significance[61]. Briefly, we found that the comparisons of cancer patients with healthy controls yielded similar AUCs for proteomics, clinical variables, and genomics (AUCs up to 0.75, 0.74, and 0.72, respectively). In contrast, metabolomics differentiated cancer patients from healthy controls with the highest AUC of only 0.64. The AUCs were considerably lower when the cancer patients were compared to non-cancer patients (Fig. 2A–C). Specifically, for proteomics, the AUC was 0.75 for all

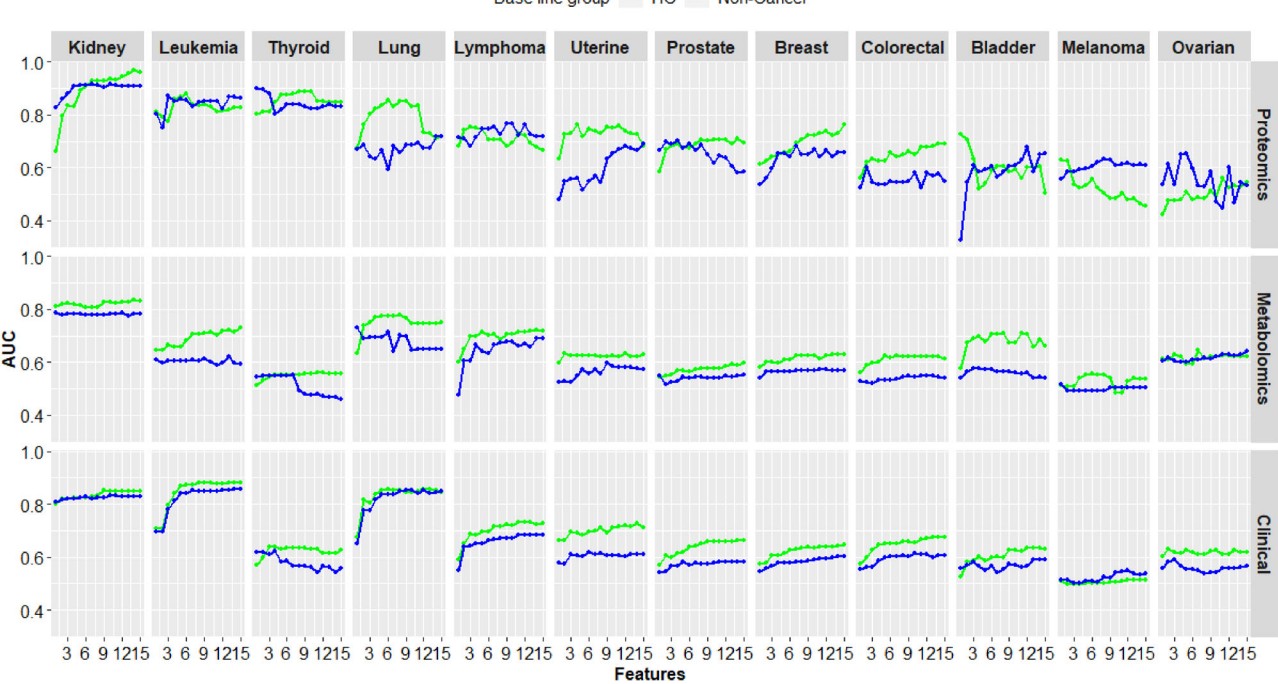

**Fig. 3 | Cancer prediction using different omics.** Line plots showing AUCs for clinical, metabolomic and proteomic variables for cancer patients compared to healthy controls (green) and patients with other diseases (blue) and the number of features for each comparison.

cancers compared to healthy controls and 0.58 compared to non-cancer patients. Similarly, when we compared cancer patients to non-cancer patients, the AUCs decreased for genomics, clinical variables, and metabolomics (0.74 to 0.56, 0.72 to 0.51, and 0.64 to 0.54, respectively). Interestingly, if the control group was completely healthy, the most important proteins to diagnose cancer were different compared to the same proteins analyzed with non-cancer controls (Fig. 2B, C). We also found that the AUCs could only be marginally improved by including more features (Fig. 2A). To increase interpretability of the results, we constructed violin plots depicting the distribution of the predictions and using *t* test, we compared the significance of the difference in the mean value of the predictions (Fig. 2D). The ROCs and most discriminative variables for every model are presented in the interactive shiny app (https://macd.shinyapps.io/CancerAtlas/).

### Analyses of individual cancers showed high AUCs for cancers in highly vascularized organs

We next investigated whether biomarkers for individual cancers could be identified. This procedure was used for cancers for which enough samples were available (at least 40 for each omics layer), namely breast, bladder, colorectal, kidney, lung, ovarian, prostate, thyroid and uterine cancers, as well as leukemia, lymphoma, and melanoma. For most of the cancers, neither the potential biomarkers from the multiomics data nor the clinical variables yielded AUC > 0.8 (Fig. 3, Supplementary Fig. 1, and Supplementary Data 4 and 5). The median (min–max) AUC for proteomics using healthy controls as a baseline group was 0.76 (0.56–0.97), while the corresponding AUCs for genomics, metabolomics, and clinical variables were 0.57 (0.45–0.62), 0.64 (0.56–0.84), and 0.67 (0.52–0.88), respectively. In support of the biological relevance of the selected proteins we performed the gene ontology term enrichment. We found that for all analyzed cancers together, regulation of cell population proliferation and of different immune cells were shared mechanisms. For individual cancers, more specific branches of these mechanisms were enriched (Supplementary Results 1, Supplementary Fig. 2, and Supplementary Data 8). Interestingly, proteomics data from four cancers yielded substantially greater AUCs than did data

from other cancers (kidney (AUC = 0.97), lung (AUC = 0.86), thyroid (AUC = 0.89), and leukemia (AUC = 0.88)). These four exceptions pointed to a hypothetical explanation for the variable predictive value of blood biomarkers: Since the four cancers are located in highly vascularized organs or in peripheral blood, this could result in greater overlap of biomarkers between local tumor cells and peripheral blood. Therefore, we hypothesized that protein biomarkers in local tumor tissue, in general, would be less variable than those in peripheral blood. In other words, protein biomarkers in local tumor tissues would be more accurate than if measured in peripheral blood, except for tumors in close proximity to peripheral blood. To our knowledge, this hypothesis has not been previously systematically examined.

### Proteomic analyses of local tumor tissue yielded high AUCs for nine cancer types

To evaluate whether the limited performance using the blood proteomics could be explained by the limited number of analyzed proteins, we explored whether using the same proteins and methods in local cancer tissues could lead to better predictions compared to those from peripheral blood. Here, we analyzed local proteomics data from breast, colon, head and neck, kidney, liver, lung, ovarian, pancreas, and uterine cancers and their respective control tissues. While only some of these cancers are available in UKBB with enough samples, the inclusion of other cancers might strengthen the results in terms of generalizability. The proteomics data included more than 10,000 proteins from 1038 patients aged 60 (49–68), 51% female, and 673 controls aged 60 (52–68), as well as 38% female (Supplementary Data 6) from the Clinical Proteomic Tumor Analysis Consortium (CPTAC). We analyzed only proteins that were measured in both CPTAC and UKBB, using the same methods as above (Supplementary Data 7). Interestingly, we found that in all cancers, only one protein was enough to differentiate tumor tissue from normal tissue, with a median AUC (min–max) of 1.00 (0.91–1.00). In addition, using up to 15 proteins yielded a median AUC (min–max) of 1.00 (0.97–1.00). The probabilities of separating tumor tissue from normal tissue using only one protein were highly significant and visually evident in the box plot (Fig. 4).

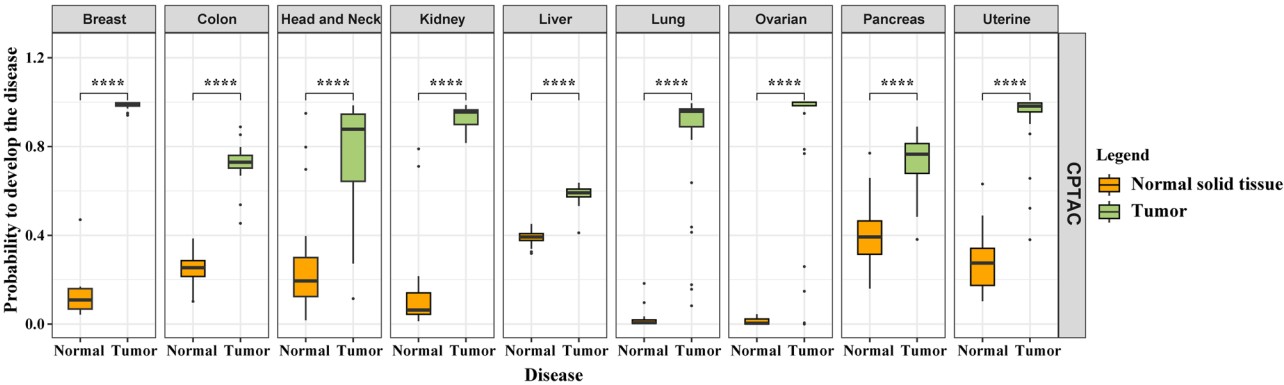

**Fig. 4 | Box plots showing the probabilities of tissue being malignant based on the use of only one protein.** The centerline corresponds to the median, the box limits to the upper and lower quartiles, the whiskers to the 1.5× interquartile range and the points to outliers. ****$P < 0.0001$. $T$ test, adjusted $P$ values.

## Discussion

Cancer is a leading cause of morbidity and mortality in most countries. In 2020, 10 million cancer deaths were estimated worldwide[25].

This has resulted in enormous investments in developing and implementing a wide variety of cancer screening methods. The aims are to reduce morbidity and mortality through early diagnosis and treatment. However, the clinical usefulness of these methods is increasingly questioned[17–20]. Reasons include overdiagnosis and the harmful effects of several diagnostic methods, such as radiation or invasive procedures. These problems could potentially be solved by tests analyzing different types of analytes in peripheral blood[8,62]. While promising results of such analyses have initially been reported, they have often proven difficult to replicate. Reasons include the potential confounding effects of the clinical settings in which the analyses are performed. AUCs can vary greatly in different clinical contexts. For example, promising results were reported when a test was used as a diagnostic complement for physicians to evaluate symptomatic patients[41]. This differs from many other screening contexts in that the patients were more likely to have cancer than was an unselected group of individuals, and that the test was part of an active diagnostic evaluation. This relates to the increasing recognition of the importance of risk-stratification in cancer screening programs[63].

In general, the highly diverse tests discussed above have in common that they are based on age and/or sex rather than additional individual-level risk factors. Examples of this include screening for cervical, breast, and prostate cancer, which are variably performed in different age groups in different countries. Another explanation is that the analytes have been identified based on analyses of individual omics layers in limited materials. This is a concern because omics analyses can include tens of thousands of variables, which require large patient cohorts to gain reproducible, statistically significant results. Moreover, any cancer screening method is complicated by the great cellular and molecular heterogeneity between different tumors. Single cell-based studies even show great heterogeneity between tumors of the same diagnosis[6]. Another complicating factor is the variable shedding of tumor analytes to blood, many of which may also be shed by normal cell types. The problem is increased by certain non-shedding tumors[64]. Indeed, our analyses showed that, in general, AUCs for analytes analyzed in tumors were considerably greater than those analyzed in blood.

Despite these many confounders and complicating factors, we hypothesized that a head-to-head comparison of proteomic, metabolomic and genomic data from peripheral blood in a large cohort could contribute to improved biomarkers to screen for some of the most prevalent cancers, either collectively or individually. We compared the cancer patients with healthy controls, as well as with patients with other diagnoses, using data from some 500,000 participants from the UKBB. Our analyses did not support that biomarkers derived from peripheral blood will be more accurate than routine clinical variables for collectively screening for cancers.

Notably, the AUCs were considerably lower when cancer patients were compared to non-cancer controls, as opposed to healthy controls. These differences could explain why many pan-cancer biomarker studies have proven difficult to replicate, and highlight the daunting challenges involved in finding reliable biomarkers for cancer screening. In contrast, relatively small numbers of proteins resulted in high AUCs for individual cancers in highly vascularized organs. This exception pointed to a potential explanation for the variable accuracy of cancer screening biomarkers in peripheral blood, namely, variable molecular overlap between local tumor cells and blood. This explanation was supported by the consistently high AUCs of proteins in local tumor tissue in nine different cancers. Thus, such proteins may be diagnostically useful if analyzed in local fluids, such as saliva or urine[65–68], however, studies with more samples are needed to further support this hypothesis. Another potential use is to analyze proteins in peripheral blood from subjects for whom screening for tumors in highly vascularized organs is needed. This is important given that comorbidities are common in many clinical settings. Future studies comparing the efficacy, usability, and cost-effectiveness of blood-based biomarkers with other modalities are needed to provide the best solutions for the patients.

A limitation of our analyses is that they were based only on 2923 proteins and 325 metabolites. Thus, higher AUCs could be reached if all proteins and metabolites were analyzed. However, our analyses of proteomics data from local tumor tissue, which included the subset of proteins analyzed in the UKBB did not support this potential, as the AUCs in tumor tissue remained considerably greater than those in blood. It is also possible that combinations of different omics biomarkers may be more accurate, but this could decrease clinical tractability because of increased cost and complexity. Other limitations were that the participants in the UKBB may not be representative of individuals of other nationalities, ethnicities, or socioeconomic conditions and that the sample size for individual cancers was variable which might have influenced the stability of the models.

To conclude, this study does not support that proteomic, metabolomic or genomic analyses of peripheral biomarkers are superior to clinical variables for collectively screening for the most common cancers. An important reason may be the variable overlap between molecular changes in local tumor tissue and peripheral blood, except for tumors in highly vascularized organs. Thus, biomarkers for such tumors in peripheral blood may prove clinically useful. We constructed a Shiny app to identify such biomarkers.

## Data availability

An interactive, web-based atlas for translational researchers to find optimal biomarkers is available at https://macd.shinyapps.io/CancerAtlas/. All data used in this study are available to access from the UK Biobank at https://www.ukbiobank.ac.uk/ for approved researchers through the UK Biobank data-access protocol. Source data is included in the Supplementary Data files.

## Code availability

All codes used in this manuscript are available at Zenodo[69].

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

## Acknowledgements

The authors are thankful to the participants and the team of the UK Biobank study. This work was supported the Swedish Cancer Society CAN 2017/411; Swedish Research Council, VR grant 2019-01165.

## Author contributions

M.S., Y.Z., D.M.A., A.F.M., and O.S. had full access to all the data in the study and took responsibility for the integrity of the data and the accuracy of the data analysis. Concept and design: M.S., M.B., and X.L. Acquisition, analysis, or interpretation of the data: all authors. Drafting of the manuscript: M.S., X. L., and M.B. Critical review of the manuscript for important intellectual content: all authors. Statistical analysis: M.S., D.M.A., A.F.M., and O.S. Obtained funding: M.B. Administrative, technical, or material support: M.S., O.S., and Y.Z. Supervision: M.B. and X.L.

## Funding

## Competing interests

M.B. is the scientific founder of Mavatar, Inc. The remaining authors declare no competing interests.
