## [Peer Review File · Communications Medicine]

This manuscript has been previously reviewed at another Nature Portfolio journal. This document only contains reviewer comments and rebuttal letters for versions considered at *Communications Medicine*.Reviewers' comments:

Reviewer #1 (Remarks to the Author):

1) General comments:

The authors present an analysis of the performance of multiomics biomarkers versus clinical variables in pan-cancer screening using data from UK Biobank participants. They report that while the overall performance of potential blood biomarkers did not outperform clinical variables for collective diagnosis, individual cancers, particularly those close to peripheral blood, showed promising results with high AUCs. They conclude that while multiomics biomarkers may not improve collective cancer diagnosis, they have potential for diagnosing individual cancers. While the manuscript's focus on the efficacy of multiomics biomarkers for pan-cancer screening is most relevant to researchers in oncology and bioinformatics, its insights into the relative performance of biomarkers and clinical variables across different cancers may be of interest to a broader audience within the journal's scope. The document is coherently structured and clearly written. However, my comments primarily relate to the lack of adjustments for multiple comparisons, the limited exploration of inherent challenges and limitations associated with blood-based cancer diagnostics and multiomics studies, such as confounders, population stratification, effects of sample size especially in subgroup analyses for certain individual cancer types, and the lack of functional validation for the identified biomarkers (see Major Comments below).

2) Specific comments:

a) Major comments for revision:

1.) Adjustment for multiple comparisons: It is not entirely clear whether the p-value significance values presented in the manuscript were adjusted for multiple hypothesis testing, and if so, what method was used. As the study involves multiple applications of statistical hypothesis testing for many different characteristics, adjustment for multiple comparisons is required. This could include the Bonferroni correction, the Benjamini-Hochberg procedure, or other procedures to control for false discovery rates. Such adjustments are essential for a multiple comparison study to control for false discovery rates.

2.) Discussion of challenges and limitations: An expanded discussion of the inherent challenges and limitations of multiomics studies is needed. This should include considerations of confounders, population stratification, and the impact of sample size, especially in subgroup analyses for specific individual cancer types. Sample sizes for some individual cancer types are significantly smaller than for others, limiting interpretation and stability of models. Further discussion of limitations is also needed regarding challenges inherent in blood-based cancer diagnostics, such as tumor heterogeneity and variability in biomarker shedding.

3.) Functional validation of biomarkers: The study lacks functional validation for the identified biomarkers. Incorporating further experimental or computational validation (e.g., using public data sources) could strengthen the findings and provide insight into the biological relevance of these biomarkers.

4.) Comparative analysis with existing methods: A more detailed comparative analysis with existing cancer screening methods could provide a clearer picture of where the multiomics approach stands in terms of efficacy, usability and cost-effectiveness.

5.) Subgroup analysis details: The manuscript should provide more details on the subgroup analyses performed for each cancer type, including the criteria for subgroup selection and the statistical methods used to ensure the reliability and statistical robustness of the results within these subgroups.

Minor comments for revision:

- 1.) Clarification of methodological details: Certain methodological details, such as the specifics of the multiomics platforms used and data preprocessing steps, need to be clarified or described in more detail for reproducibility and to allow readers to fully understand the approach.
- 2.) Figures: Some figures are difficult to interpret due to very small font sizes, especially for axis labels or legends.
- 3.) References to recent literature: The manuscript could benefit from references to recent studies in the field of multiomics and cancer screening to contextualize its contributions within the current research landscape.

Reviewer #2 (Remarks to the Author):

In this manuscript, Smelik et al. address a relevant and still open question how blood-based parameters can be used to improve cancer screening methods and diagnosis of individual tumor types. While this is a relevant research question that needs to be addressed by comprehensive (computational) approaches, the present study is not able to address this adequately.

In general, the manuscript provides a very superficial description of the data and results. Interpretations of findings are not sufficiently supported by the analyses performed and their results. It remains unclear if features identified here are indeed associated with the underlying diseases and to what extent. Many features that are identified to have biomarker potential in individual cancers are quite general, not specific to the entity (BMI, hand grip strength, etc.).

Have the authors looked into subgroups of cancer? Merging samples based on entity alone is not quite meaningful. The fact that AUC values are lower when testing all cancers (in comparison to individual cancer entities) might be an effect of reduced sample size rather than specificity for the underlying disease. Have authors evaluated the features in multivariate testing?

The stability of results can not be evaluated, e.g. as it remains unclear how training and test data sets have been used (e.g. by repeated sampling) and how results have been validated independently. The time point of diagnosis (relative to sampling) is unclear, yet highly important to evaluate biomarkers for detection! Further, it is unclear if non-cancer patients have been diagnosed with cancer after sampling!

Several conceptual steps regarding data/feature/method selection remain unclear as they are not well described. E.g. unclear why PRSs have been used, this is not well introduced. Also, why has CPTAC data been selected as control here? Such information must be given to evaluate the findings.

In general, it remains unclear how this study describes novel findings over previous publications. This should be clearly addressed in an improved version of the manuscript.

Individual remarks:

- line 52: "UKBB" is not introduced
- l. 55: "healthy controls, as well as non-cancer controls": unclear at this point of the manuscript
- l. 58: "collectively and individually": unclear wording, this should be rephrased to describe what is meant here
- l. 69 "local body fluids, such as saliva or urine": not supported by any data or finding in this study
- l. 79: "90 million genetic variants": have genetic variants also been merged, e.g. by functional consequence or by gene? is the functional consequence or pathologic assessment of variants included in any way? also, in the Methods section the authors describe 900 million variants, not 90!
- l. 80: "biomarker potential": potential for what? please be precise
- l. 81: "we repeated the analyses for individual cancers": based on the previous analysis across all

cancers or independent of this? again, this is unclear

- l. 102: "PRS": not introduced yet
- l. 103: why ridge regression? have the authors test also other approaches?
- l. 104: "three genetic principal components": this is not introduced, why PCs? please elaborate
- l. 105: "ROC curve as well as AUC values": this is not as is shown in eTables 2 and 3! moreover, how have they been calculated? this is unclear but essential to evaluate the findings
- l. 107: "aimed to achieve AUCs above 0.8": how did the authors "aimed" at this? that's confusing, did it influence the results? should be rephrased!
- l. 109: "genomics": not shown in the figures
- l. 109: "Specifically, for proteomics": this should be described for all data levels together, not separated for proteomics and others
- l. 117: "improved": the values described here already show the "improvement", this needs to be rephrased
- l. 133: "enough": why are 40 considered "enough"?
- l. 134: "proteomics": how many for other data layers?
- l. 135: "high AUCs": consider rephrasing, i.e. what is "high"? >0.8 ? this cutoff could be shown in the figures as a goal to reach
- l. 143: what about lymphomas?
- l. 144: overlap of what? please describe findings and hypotheses in a relevant detail!
- l. 201-203: this is not what has been tested here and thus does not support this hypothesis
- l. 207-209: this is in contrast to what the authors have described previously (l. 193-195)
- figure 2: data shown in figures 2B(left part),C,D is not discussed in the text

General response: We thank reviewers for the insightful comments and suggestions. We provide the revised manuscript and the revised supplementary files, in which we addressed the issues highlighted by reviewers as explained below in more detail.

Reviewer #1 (Remarks to the Author):

1) General comments:

The authors present an analysis of the performance of multiomics biomarkers versus clinical variables in pan-cancer screening using data from UK Biobank participants. They report that while the overall performance of potential blood biomarkers did not outperform clinical variables for collective diagnosis, individual cancers, particularly those close to peripheral blood, showed promising results with high AUCs. They conclude that while multiomics biomarkers may not improve collective cancer diagnosis, they have potential for diagnosing individual cancers.

While the manuscript's focus on the efficacy of multiomics biomarkers for pan-cancer screening is most relevant to researchers in oncology and bioinformatics, its insights into the relative performance of biomarkers and clinical variables across different cancers may be of interest to a broader audience within the journal's scope. The document is coherently structured and clearly written. However, my comments primarily relate to the lack of adjustments for multiple comparisons, the limited exploration of inherent challenges and limitations associated with blood-based cancer diagnostics and multiomics studies, such as confounders, population stratification, effects of sample size especially in subgroup analyses for certain individual cancer types, and the lack of functional validation for the identified biomarkers (see Major Comments below).

2) Specific comments:

a) Major comments for revision:

1.) Adjustment for multiple comparisons: It is not entirely clear whether the p-value significance values presented in the manuscript were adjusted for multiple hypothesis testing, and if so, what method was used. As the study involves multiple applications of statistical hypothesis testing for many different characteristics, adjustment for multiple comparisons is required. This could include the Bonferroni correction, the Benjamini-Hochberg procedure,

or other procedures to control false discovery rates. Such adjustments are essential for a multiple comparison study to control false discovery rates.

Response: The adjustment for multiple comparisons was used in all cases where p values are presented. We updated the supplementary methods (lines 103-104) to clearly state that the Holm method was used for adjustment for multiple comparisons.

2.) Discussion of challenges and limitations: An expanded discussion of the inherent challenges and limitations of multiomics studies is needed. This should include considerations of confounders, population stratification, and the impact of sample size, especially in subgroup analyses for specific individual cancer types. Sample sizes for some individual cancer types are significantly smaller than for others, limiting interpretation and stability of models. Further discussion of limitations is also needed regarding challenges inherent in blood-based cancer diagnostics, such as tumor heterogeneity and variability in biomarker shedding.

Response: We thank the reviewer for these comments, which have led us to expand the discussion (lines 214-229) in accordance with those comments. Specifically, we have discussed the confounding effects of 1) different clinical settings, including if the test was used as a diagnostic complement by a physician to evaluate a symptomatic patient, or in an unselected population; 2) variable implementation of risk-stratification, which in general does not include evaluation of individual risk factors; 3) variable shedding of tumor analytes; 4) the great cellular and molecular heterogeneity between and within different tumor diagnoses (even within the latter there may be great heterogeneity), all of which may confound or complicate cancer screening.

3.) Functional validation of biomarkers: The study lacks functional validation for the identified biomarkers. Incorporating further experimental or computational validation (e.g., using public data sources) could strengthen the findings and provide insight into the biological relevance of these biomarkers.

Response: To strengthen the findings and provide biological relevance, we performed gene ontology term enrichment analysis of 15 most disease relevant proteins. In the revised results, we have added the summary of the result to the manuscript (lines 159-163) and detailed description of the results into the supplement (eResults 1). Shortly, for each cancer, we

selected the five most significant biological processes based on gene ontology database and constructed a heatmap to visualize the results. For all analyzed cancers together, regulation of cell population proliferation and of different immune cells were shared mechanisms. For individual cancers, more specific branches of these mechanisms were enriched. The list of all significantly enriched GO terms for all cancers can be newly found in eTable7 and the heatmap representing the most significant GO terms is available as new eFigure 2.

4.) Comparative analysis with existing methods: A more detailed comparative analysis with existing cancer screening methods could provide a clearer picture of where the multiomics approach stands in terms of efficacy, usability and cost-effectiveness.

Response: We extensively revised the introduction to respond to this comment. (lines 34-37, 52-60) We summarize the cancer screening results from other publications across different cancers and using different modalities, such as imaging technologies or genome-wide mutations. We also refer to the recent review articles that summarize the current state of cancer screening for different cancers or different modalities. To our knowledge, there are no reviews that summarize multiple cancers as well as multiple modalities. However, we believe that this manuscript bridges this gap by providing systematic insights of how different omics layers and clinical variables may be used in a range of different cancers. Additionally, we revised the discussion (lines 249-251) to suggest future studies to compare blood-based biomarkers with other modalities.

5.) Subgroup analysis details: The manuscript should provide more details on the subgroup analyses performed for each cancer type, including the criteria for subgroup selection and the statistical methods used to ensure the reliability and statistical robustness of the results within these subgroups.

Response: In the updated version of the manuscript, we clarified the reasoning behind grouping different cancer types and for selecting the particular groups (lines 285-286). Shortly, the grouping was made based on the stratification provided by UKBB (reference 63) and we selected all cancers that had enough samples to keep statistical power of the analysis. The reliability of the statistical robustness was based on selecting cancer with enough samples as well as selecting the methods that are appropriate for both larger and lower sample size (such as extremely randomized trees or logistic regression). Despite this, we

acknowledge in the revised discussion (lines 259-262) that the results might be influenced by the variable number of samples for different cancers.

Minor comments for revision:

1.) Clarification of methodological details: Certain methodological details, such as the specifics of the multiomics platforms used and data preprocessing steps, need to be clarified or described in more detail for reproducibility and to allow readers to fully understand the approach.

Response: The specifics of the multiomics platforms are available in the supplementary file in eMethods 1. The data creation was all done by the UKBB project, and we provide references to their original publications where they describe everything in detail. The methodological details about our processing of the data are provided in eMethods 2-4. Moreover, we provide all codes necessary to either fully reproduce the results or clarify some minor details about the packages and methods that were used in the manuscript. We clarified the methodology at lines 290-292 and 301-304 in the revised manuscript and at lines 60-63 in the eMethods 2 and lines 73-80 in eMethods 3.

2.) Figures: Some figures are difficult to interpret due to very small font sizes, especially for axis labels or legends.

Response: We updated the figures to increase the readability of the labels and legends

3.) References to recent literature: The manuscript could benefit from references to recent studies in the field of multiomics and cancer screening to contextualize its contributions within the current research landscape.

Response: In the revised introduction, we have added new references both in the multiomics field and cancer screening. These were mainly related to the major comment 4. Additionally, in the revised discussion, we discussed how our results relate to the results presented by the previous studies (lines 211-227)

Reviewer #2 (Remarks to the Author):

In this manuscript, Smelik et al. address a relevant and still open question how blood-based parameters can be used to improve cancer screening methods and diagnosis of individual tumor types. While this is a relevant research question that needs to be addressed by comprehensive (computational) approaches, the present study is not able to address this adequately.

In general, the manuscript provides a very superficial description of the data and results. Interpretations of findings are not sufficiently supported by the analyses performed and their results. It remains unclear if features identified here are indeed associated with the underlying diseases and to what extent. Many features that are identified to have biomarker potential in individual cancers are quite general, not specific to the entity (BMI, hand grip strength, etc.).

Have the authors looked into subgroups of cancer? Merging samples based on entity alone is not quite meaningful. The fact that AUC values are lower when testing all cancers (in comparison to individual cancer entities) might be an effect of reduced sample size rather than specificity for the underlying disease. Have authors evaluated the features in multivariate testing?

Response: We agree that the analysis of different subgroups of cancers would be relevant. However, to do this systematically for multiple cancers, with sufficient statistical power would require cohorts of much larger size than the UK biobank data, which comprises more than 500,000 participants. In the updated manuscript, we have revised the discussion (lines 259-262) to acknowledge the limitations regarding variable sample size between different cancers. We used the extremely randomized trees method to evaluate the importance of different features as explained in the manuscript at lines 325-327 and supplement methods at lines 88-89.

The stability of results can not be evaluated, e.g. as it remains unclear how training and test data sets have been used (e.g. by repeated sampling) and how results have been validated independently. The time point of diagnosis (relative to sampling) is unclear, yet highly

important to evaluate biomarkers for detection! Further, it is unclear if non-cancer patients have been diagnosed with cancer after sampling!

Response: The datasets were processed in the following way (As an example, we will use lung cancer but “lung” could be replaced by any other cancer or even all cancer patients as the computational pipeline was the same for each scenario):

- 1) We selected all lung cancer patients from UK biobank, that were diagnosed at the time of sampling.
- 2) We selected non-cancer patients and healthy controls and we computationally selected those that corresponded to the lung cancer patients by sex and age. The non-cancer patients were selected as those that did not suffer from lung cancer until the end of the follow up period (31st of October 2022)
- 3) We divided the data between train and test
- 4) We constructed the pipeline using only train dataset
- 5) We tested the performance of the pipeline using the previously unseen test dataset.

The sampling was done at the baseline assessments between 2006 and 2010.

We updated the methodology to clarify the processing of the data as explained above at lines 290-292 in main file and lines 60-63 in supplementary methods.

Several conceptual steps regarding data/feature/method selection remain unclear as they are not well described. E.g. unclear why PRSs have been used, this is not well introduced. Also, why has CPTAC data been selected as control here? Such information must be given to evaluate the findings.

Response: The aim of our work was to compare all relevant blood-related data sources. It has been shown that the PRS can provide an estimate of an individual’s risk of developing certain diseases, including cancers. PRS take into account the cumulative effect of many genetic variants, each contributing a small amount of overall risk for the disease and thus we wanted to analyze, how well they differentiate cancer patients from controls compared to other omics layers and clinical variables. The CPTAC data were used to evaluate whether the limited AUC values that we obtained for blood-based biomarkers could be explained by lack of relevant biomarkers, or because of molecular differences between the local tissue and blood.

We found that using the same proteins, we were able to obtain substantially higher AUC using the proteins from local tissue, which agrees with molecular changes in blood only partially reflecting those in tumor tissues. The motivation for using PRS can be found in lines 301-304 of the revised manuscript and the motivation to use CPTAC results is available at lines 178. We also updated the eMethods 3 (lines 68-69, 73-80) to explain PRS in more detail.

In general, it remains unclear how this study describes novel findings over previous publications. This should be clearly addressed in an improved version of the manuscript.

Response: The introduction has been extensively revised to clarify the novel aspects of our manuscript: Briefly, Proteomic analysis of several diseases, including cancers, based on UKBB have identified promising biomarkers linked to both mortality and cancer incidence. The limitations of these studies included that they were focused on incident cases, in other words patients diagnosed after blood sampling. These results led us to construct models to differentiate already diagnosed patients from healthy controls, as well as from patients with non-cancer diseases as controls, and to compare their performance with the predictive models based on tissue proteomics. Moreover, we systematically compared the predictive accuracy of different omics layers.

Individual remarks:

- line 52: "UKBB" is not introduced

Response: We have revised the manuscript to make sure that all abbreviations, including UKBB are introduced properly

- l. 55: "healthy controls, as well as non-cancer controls": unclear at this point of the manuscript

Response: We have introduced term non-cancer controls to clarify this sentence for the reader. (lines 66-67)

- l. 58: "collectively and individually": unclear wording, this should be rephrased to describe what is meant here (lines 68-70)

Response: We have rephrased the sentence in the revised manuscript

- 1. 69 "local body fluids, such as saliva or urine": not supported by any data or finding in this study

Response: We removed this part from the introduction. We motivate the other claims about the local body fluids such as saliva or urine based on the previous literature and in the discussion, we encourage future studies regarding local body fluids.

- 1. 79: "90 million genetic variants": have genetic variants also been merged, e.g. by functional consequence or by gene? is the functional consequence or pathologic assesment of variants included in any way? also, in the Methods section the authors describe 900 million variants, not 90!

Response: In the revised manuscript we corrected the sentence about 900 million variants to clarify that the dataset included 90 million variants (line 303). The genetic variants used in this study were those that have been published in the cancer-specific polygenic risk scores reported in the PGS catalog. For each cancer specific PRS, each reported genetic variant was weighed by its reported effect size and used to calculate the individual risk score for this specific cancer. For the combined PRS, the genetic variants reported to be associated with each cancer were grouped together and used to calculate the individual combined risk score. There was no selection of genetic variants based on functional consequence or based on the genes these genetic variants were mapped to. We clarified these in the method and eMethods 3.

- 1. 80: "biomarker potential": potential for what? please be precise

Response: We revised the manuscript to clarify this, now in line 94: "Using machine learning, we first identified the potential diagnostic biomarkers using these variables in all cancers available in the UKBB cohort compared to controls"

- 1. 81: "we repeated the analyses for individual cancers": based on the previous analysis across all cancers or independent of this? again, this is unclear

Response: The same data processing pipeline was used both in case of all cancers and cancers individually. The training, however, was done separately for each case. We clarified our approach in the revised manuscript (lines 94-96).

- 1. 102: "PRS": not introduced yet

Response: We have revised the manuscript to make sure that all abbreviations, including PRS are introduced properly (line 117)

- 1. 103: why ridge regression? have the authors test also other approaches?

Response: The motivation is provided in the Supplementary file (lines 90-97). Briefly, logistic regression with the L2 penalty was preferred because of its interpretability, which is crucial in clinical research. Logistic regression also facilitates the assessment of each feature's positive or negative association with the disease, aligning with the interpretative needs of clinicians familiar with this method.

- 1. 104: "three genetic principal components": this is not introduced, why PCs? please elaborate

Response: We have introduced the motivation for using PCs in the revised manuscript (lines 119-121). Shortly, the motivation was to limit the effects of possible disease unrelated confounders in our models.

- 1. 105: "ROC curve as well as AUC values": this is not was is shown in eTables 2 and 3! moreover, how have they been calculated? this is unclear but essential to evaluate the findings

Response: We updated the references to the ROC curves (in shiny app) and AUCs (eTables 3,4). We updated the supplementary file to provide information about how the AUCs were computed (lines 100-101)

- 1. 107: "aimed to achieve AUCs above 0.8": how did the authors "aimed" at this? thats confusing, did it influence the results? should be rephrased!

Response: In the revised manuscript (line 124) we have clarified that this cut-off was based on the previous literature, specifically reference 53. The values of 0.8 did not influence our results, we include this value to provide the information about what levels of AUC are generally considered to be high and to provide reference for readers if they are interested to find out more.

- 1. 109: "genomics": not shown in the figures

Response: That is because we used PRS for genomics instead of variable number of features, thus only Fig 2D is and eFig 1 are relevant for genomics

- 1. 109: "Specifically, for proteomics": this should be described for all data levels together, not separated for proteomics and others

Response: All the results are present on figure 2A and can be specifically analyzed using the Shiny app based on what exactly the reader is interested in. Moreover, the values for the other layers are available in the next sentence.

- 1. 117: "improved": the values described here already show the "improvement", this needs to be rephrased

Response: The improvement relates to using more features in the model (for example 15 instead of 2). The values presented in the line above correspond the usage of different control group (HC or non-cancer). We restructured the results (lines 131-135) to make clear that those two sentences are not related.

- 1. 133: "enough": why are 40 considered "enough"?

Response: We chose to include cancers with at least 40 cases to ensure that our analysis had adequate statistical power. Additionally, this threshold allowed us to include a diverse range of cancers, enhancing the generalizability of our findings across different cancer types. It also mitigated the risk of overfitting the machine learning models, which can occur when analyzing very small sample sizes. We believe that this value provides a balance between inclusivity and the robustness of our results.

- 1. 134: "proteomics": how many for other data layers?

Response: Generally, there was more data, as metabolomics, genetics and clinical variables were available for more patients. The exact numbers are available in eTable 2.

- 1. 135: "high AUCs": consider rephrasing, i.e. what is "high"? >0.8 ? this cutoff could be shown in the figures as a goal to reach

Response: We rephrased the sentence (lines 154-156) to clarify that by high, we mean 0.8. The figures include lines corresponding to $AUC = 0.8$ which could be used to estimate the exact values directly from the figures.

- 1. 143: what about lymphomas?

Response: The respective AUC for lymphomas was 0.75. Thus, it was lower than 0.8 and considerably lower than other cancers that we mentioned. The exact value for lymphomas as well as other cancers can be found in eTable 3.

- 1. 144: overlap of what? please describe findings and hypotheses in a relevant detail!

Response: In the revised manuscript (lines 168-169), we clarified that by overlap, we mean the overlap of biomarkers between local tumor tissue and peripheral blood.

- 1. 201-203: this is not what has been tested here and thus does not support this hypothesis

Response: We were testing whether it is possible to achieve higher AUCs using the proteomics from local tumor tissue compared to the same proteins coming from blood. We found (lines 181-184) that in all cancers, only one protein was enough to differentiate tumor tissue from normal tissue, with a median AUC (min-max) of 1.00 (0.91-1.00). Additionally, using up to 15 proteins yielded a median AUC (min-max) of 1.00 (0.97-1.00). In the revised manuscript, we clarified the statement from 'consistently high diagnostic accuracy' to 'consistently high AUCs' (243-245) as indeed, we compared the AUCs instead of accuracy.

- l. 207-209: this is in contrast to what the authors have described previously (l. 193-195)

Response: We removed the statement from the discussion.

- figure 2: data shown in figures 2B(left part),C,D is not discussed in the text

Response: We updated the text in lines 133-138 to clarify the importance of the figures

Reviewers' comments:

Reviewer #1 (Remarks to the Author):

The authors have addressed all the main comments and their revisions have improved the manuscript.

Reviewer #2 (Remarks to the Author):

The revised manuscript has substantially improved and is much clearer to read than the original version. The authors improved the presentation of their findings as well as their conclusions. Overall the manuscript is more concise and focussing on the aspects this study is able to address. Still, individual adjustments are necessary:

- line 35-37: this is a very technical description. the authors should rather focus on the meaning/interpretation of findings, not at exact results.
- l. 76: "high area under the receiver operating characteristic (ROC) curves (AUCs) were found." - indicating what? again, this is very technical and the authors should explain what this means in terms of interpretation
- l. 95: "all cancers" - please describe in the text how many cancers are available, and provide sample numbers per entity
- l. 98: "healthy controls" - the authors still do not describe how long after the inclusion into UKBB, these individuals have not been diagnosed with any cancer. this should be given, e.g. as median +min/max or +IQR to allow for interpretation of the results
- l. 132: "The most relevant proteins differed greatly based on the control group that was used" - this is a difficult sentence, the authors should consider rephrasing it to improve readability
- l. 150: "with an AUC >0.8" - delete: so far no biomarkers with AUC>0.8 have been described. thus, this doesnt make sense here.
- l. 151: "This procedure was used for cancers" - how many cancers entities are analyzed here? please provide numbers
- l. 152: "40 for proteomics" - how many for other omics layers?
- l. 171/Figure 3: ordering the plots as based on (average) AUC would increase readability and improve the illustration of the results. Also Omics layers should be ordered accordingly, i.e. showing Proteomics at the top
- l. 176: "local proteomics" - please explain differences to the previously described analysis in more detail to improve the understanding
- l. 176: "nine cancers" - please explain this selection, what is the overlap with the UKBB cancer entities? how many samples per entity?
- l. 185: "only one protein" - what are these proteins? have they been identified using the UKBB data as well? how does the overlap look like, is there any?
- l. 209: "these" - this is a new paragraph, so please spell out what is meant here
- l. 214: "large matgerials" - what is meant with this?
- l. 216-217: this is not well supported by the results. either show corresponding results in detail or remove this sentence
- l. 231: "as opposed to compared to healthy controls" - please rephrase sentence
- l. 283: "31st of October" - year is missing. also please provide the median (+min/max or IQR) duration

The revised manuscript has substantially improved and is much clearer to read than the original version. The authors improved the presentation of their findings as well as their conclusions. Overall the manuscript is more concise and focussing on the aspects this study is able to address. Still, individual adjustments are necessary:

Response: We thank the reviewer for all suggestions. We incorporated them into the updated version of the manuscript as explained in detail below:

- line 35-37: this is a very technical description. the authors should rather focus on the meaning/interpretation of findings, not at exact results.

Response: In the revised manuscript we have modified this sentence to make it more accessible (lines 35-37):

“The performance of these technologies showed substantial variations depending on the study characteristics.”

- l. 76: "high area under the receiver operating characteristic (ROC) curves (AUCs) were found." - indicating what? again, this is very technical and the authors should explain what this means in terms of interpretation

Response: We have modified the paragraph to make it more accessible (lines 74-77):

“For most cancers we found limited diagnostic potential for all omics layers as well as clinical variables. However, for some cancers in highly vascularized organs, such as the kidney and the thyroid, a greater diagnostic potential was found as depicted by high area under the receiver operating characteristic (ROC) curves (AUCs).”

- l. 95: "all cancers" - please describe in the text how many cancers are available, and provide sample numbers per entity

Response: In the revised results, we have clarified that 97 cancers were analyzed. We also include a new supplementary table (eTable 1 in Supplement 2) that provides the number of patients per each cancer (lines 92-98):

“We identified 32,260 cancer patients with 86 cancers (eTable 1 in Supplement 2). Their median [IQR] age was 62 [57-66] years, 59% of whom were female. The dataset included 2,923 proteins, 325 metabolites, 90 million genetic variants and 45 clinical variables (eTable 2 in Supplement 2).

Using machine learning, we first identified the potential diagnostic biomarkers using these variables collectively in all cancers (ICD-10 code starting with C) available in the UKBB cohort compared to controls.”

- l. 98: "healthy controls" - the authors still do not describe how long after the inclusion into UKBB,

these individuals have not been diagnosed with any cancer. this should be given, e.g. as median +min/max or +IQR to allow for interpretation of the results

Response: In the revised results, we added the information about the follow up period date and IQR (lines 103-104):

“The follow up period of the patients was until the 31st of October 2022, thus the median (IQR) duration of follow up was 5007 (481) days.”

- l. 132: "The most relevant proteins differed greatly based on the control group that was used" - this is a difficult sentence, the authors should consider rephrasing it to improve readability

Response: We rephrased the sentence to improve the readability of this section (lines 135-137):

“Interestingly, if the control group was completely healthy, the most important proteins to diagnose cancer were different compared to the same proteins analyzed with non-cancer controls.”

- l. 150: "with an AUC >0.8" - delete: so far no biomarkers with AUC>0.8 have been described. thus, this doesnt make sense here.

Response: We removed this part of the sentence and rephrased it to improve the readability (line 154):

“We next investigated whether biomarkers for individual cancers could be identified.”

- l. 151: "This procedure was used for cancers" - how many cancers entities are analyzed here? please provide numbers

Response: In order to clarify the results, we newly provide the names of all cancers which were analyzed (lines 154-157):

“This procedure was used for cancers for which enough samples were available (at least 40 for each omics layer), namely breast, bladder, colorectal, kidney, lung, ovarian, prostate, thyroid and uterine cancers, as well as leukemia, lymphoma and melanoma.”

- l. 152: "40 for proteomics" - how many for other omics layers?

Response: As described in the response to the comment above, the modified sentence clarifies that it was at least 40 patients for each omics layer (lines 155-156).

- l. 171/Figure 3: ordering the plots as based on (average) AUC would increase readability and improve the illustration of the results. Also Omics layers should be ordered accordingly, i.e. showing Proteomics at the top

Response: We reorganized the figure as instructed by the reviewer.

- l. 176: "local proteomics" - please explain differences to the previously described analysis in more detail to improve the understanding

Response: Since this and the following two comments are closely related we provide a coherent response after the third comment below:

- I. 176: "nine cancers" - please explain this selection, what is the overlap with the UKBB cancer entities? how many samples per entity?

Response: Please see the coherent response after the next comment. The nine cancers were available in CPTAC. We listed the cancers in the revised version of the results. The number of samples for each cancer is available as eTable 6 in supplement 2.

- I. 185: "only one protein" - what are these proteins? have they been identified using the UKBB data as well? how does the overlap look like, is there any?

Response: We listed the cancers in the revised version of the results. The number of samples for each cancer is available as eTable 6 in supplement 2. In the revised version of the results, we also provided rationale for the analysis, the rationale for the proteins used in the analysis and the rationale for why we are also analyzing cancers that were not available in the UKBB (lines 169-177 and 183-194):

We first describe the rationale for analyzing local tumor biomarkers in the end of the results section entitled **“Analyses of individual cancers showed high AUCs for cancers in highly vascularized organs”**:

“These four exceptions pointed to a hypothetical explanation for the variable predictive value of blood biomarkers: Since the four cancers are located in highly vascularized organs or in peripheral blood, this could result in greater overlap of biomarkers between local tumor cells and peripheral blood. Therefore, we hypothesized that protein biomarkers in local tumor tissue, in general, would be less variable than those in peripheral blood. In other words, protein biomarkers in local tumor tissues would be more accurate than if measured in peripheral blood, except for tumors in close proximity to peripheral blood. To our knowledge, this hypothesis has not been previously systematically examined.”

The resulting analyses supported the hypothesis. However, another explanation for lower diagnostic accuracy could be that fewer proteins were analyzed in peripheral blood than in local tumor tissues. This led us to additional analyses to examine this explanation, which also responds to the comment about why cancers that were not available in UKBB were analyzed in CPTAC:

“To evaluate whether the limited performance using the blood proteomics could be explained by the limited number of analyzed proteins, we explored whether using the same proteins and methods in local cancer tissues could lead to better predictions compared to those from peripheral blood. Here, we analyzed local proteomics data from breast, colon, head and neck, kidney, liver, lung, ovarian, pancreas and uterine cancers and their respective control tissues. While only some of these cancers are available in UKBB with enough samples, the inclusion of other cancers might strengthen the results in terms of generalizability. The proteomics data included more than 10,000 proteins from 1,038 patients aged 60 [49-68], 51% female, and 673 controls aged 60 [52-68], as well as 38% female (eTable 6 in supplement 2) from the Clinical Proteomic Tumor Analysis Consortium (CPTAC). We analyzed only proteins that were measured in both CPTAC and UKBB, using the same methods as above (eTable 7 in supplement 2).”

- I. 209: "these" - this is a new paragraph, so please spell out what is meant here

Response: In the revised manuscript we clarified that we meant (lines 221-222):

“In general, the highly diverse tests discussed above have in common that they are based on age and/or sex rather than additional individual level risk factors. “

- I. 214: "large matgerials" - what is meant with this?

Response: We clarified that large patient cohorts are necessary for such analyses (lines 225-227):

“This is a concern because omics analyses can include tens of thousands of variables, which require large patient cohorts to gain reproducible, statistically significant results.”

- I. 216-217: this is not well supported by the results. either show corresponding results in detail or remove this sentence

Response: We removed the sentence as suggested by the reviewer

- I. 231: "as opposed to compared to healthy controls" - please rephrase sentence

Response: We rephrased the sentence as suggested by the reviewer (lines 241-242):

“Notably, the AUCs were considerably lower when cancer patients were compared to non-cancer controls, as opposed to healthy controls.”

- I. 283: 31st of October" - year is missing. also please provide the median (+min/max or IQR) duration

Response: We added the year in the method section and provide IQR in the revised results section (lines 294-295):

“The follow up period of the patients was until the 31st of October 2022.”

Reviewers' comments:

Reviewer #2 (Remarks to the Author):

I have one comment, which the authors haven't addressed yet:

"only one protein" - what are these proteins? have they been identified using the UKBB data as well? how does the overlap look like, is there any?

Would it be possible to give the names of these proteins?

Comment 1) I have one comment, which the authors haven't addressed yet:

Response: We thank the reviewer for this comment, and we provide the clarifications below to each question posed by the reviewer.

Question 1: "only one protein" - what are these proteins?

Response: These are proteins coming from CPTAC from different tumor tissues compared to healthy tissues. These proteins were included if they 1) were found in both CPTAC and UKBB and 2) if only one protein yielded very high AUCs using the test dataset. This is described in the results as follows:

“We analyzed only proteins that were measured in both CPTAC and UKBB, using the same methods as above (eTable 7 in supplement 2). Interestingly, we found that in all cancers, only one protein was enough to differentiate tumor tissue from normal tissue, with a median AUC (min–max) of 1.00 (0.91-1.00). Additionally, using up to 15 proteins yielded a median AUC (min–max) of 1.00 (0.97-1.00). The probabilities of separating tumor tissue from normal tissue using only one protein were highly significant and visually evident in the box plot (Figure 4).”

Question 2) have they been identified using the UKBB data as well?

Response: Yes, as described in the previous response all these proteins were identified in UKBB as well.

Question 3) How does the overlap look like, is there any?

Response: Yes, exactly the same proteins in CPTAC and UKBB were studied.

Question 4) Would it be possible to give the names of these proteins?

Response: Yes, they are available in the interactive atlas: [Multiomics Atlas of Biomarkers for cancer diagnostics \(shinyapps.io\)](https://shinyapps.io/multiomics-atlas/). As an example, you can get the names of these proteins in uterine cancer in local cancer tissue by these settings: Select Uterine in the disease category, CPTAC in the Omics category and #Features to 1. This shows the protein with the highest AUC, in this specific case AHNAK.

REVIEWERS' COMMENTS:

Reviewer #2 (Remarks to the Author):

The authors have addressed the questions, however, the names of the proteins that are sufficient to identify tumor samples are still not provided in the manuscript. This might also be realised in a table format in the supplement. However, this is a very interesting finding and should be provided. Pointing to the web resource is not sufficient in my opinion. Other than this, I do not have any further comments.